# Supplementation with Fermented Feedstuff Enhances Orexin Expression and Secretion Associated with Increased Feed Intake and Weight Gain in Weaned Pigs

**DOI:** 10.3390/ani12101329

**Published:** 2022-05-23

**Authors:** Yang Lu, Ruiyang Zhang, Hulong Lei, Yiqiong Hang, Huiqin Xue, Xuan Cai, Yonghong Lu

**Affiliations:** 1Institute of Animal Husbandry and Veterinary Science, Shanghai Academy of Agricultural Sciences, Shanghai 201106, China; leihulong@saas.sh.cn (H.L.); mjfungi@126.com (Y.H.); xuehuiqin163@163.com (H.X.); caixuan1985911@163.com (X.C.); 2College of Animal Science and Veterinary Medicine, Shenyang Agricultural University, Shenyang 110866, China; zhangruiyang@syau.edu.cn

**Keywords:** fermented feedstuff, growth performance, orexin, pigs

## Abstract

**Simple Summary:**

In weaned pigs, finding a strategy to improve the health status while also enhancing the growth performance of piglets under weaning stress through nutritional measures is an important issue that needs to be solved. Supplementation with fermented feedstuff improves the feed intake and growth performance of weaned pigs, but the exact mechanism behind this remains unclear. Hence, the present study evaluated the effects of fermented feedstuff on the performance and gastrointestinal hormones involved in feed intake and growth in weaned pigs. The results of our study showed that dietary supplementation with fermented feedstuff improved the growth performance of weaned pigs by increasing orexin, IGF-1, and IGFR levels.

**Abstract:**

The health status of weaned pigs is crucial for their subsequent growth performance. Supplementation with fermented feedstuff is able to improve the feed intake and growth of weaned pigs; however, the exact mechanism behind this is not clear. Hence, in the present study a total of 320 Duroc × Landrace × Yorkshire weaned pigs were selected and allocated to the following two groups: unfermented diet group (UFD) and fermented diet group (FD). The experimental period lasted 21 days. At the end of the experiment, feces, blood, and gastrointestinal tissue samples (including the stomach, jejunum, and ileum) were collected and used for further analysis. The results of growth performance suggested that the FD group had significantly increased (*p* < 0.05) average daily feed intake (ADFI) and average daily gain (ADG) during the first week, during the last two weeks, and over the entire three-week period compared with the UFD group. The results of the apparent nutrient digestibility of pigs showed that, compared with the UFD group, the FD group showed increased phosphorus (*p* < 0.05) and CP (*p* < 0.1) digestibility. There were no significant differences in the serum biochemical parameters between the UFD and FD groups. Moreover, our results showed that the FD group showed significantly increased gene expression of SGLT1 and PepT1 in the jejunum (*p* < 0.05). Compared with the UFD group, the FD group showed an increased (*p* < 0.05) serum orexin level and prepro-orexin (PPOX) expression in the gastric fundus, jejunum, and ileum mucosa and increased IGF-1 and IGFR expression in the jejunum. Collectively, these results indicated that supplementation with fermented feedstuff in the diet effectively enhanced the feed intake and growth of weaned pigs and that this may have been caused by the increased orexin, IGF-1, and IGFR serum levels.

## 1. Introduction

Given animal productivity and economic profits, the growth performance and health status of weaning piglets are issues of great concern in the swine industry. Weaning exposes piglets to various stress factors, leading to growth retardation and diarrhea due to low feed intake and disorders involving gastrointestinal function. Hence, it is imperative to explore healthier nutritional measures that can improve the growth performance of weaned pigs.

The fermentation of feed may be advisable, especially in the early growth stage of pigs, from a health point of view. Microbial activities during fermentation partially degrade antinutritional factors in the feed, such as cell wall components and phytates, and break down the large molecule nutrients, such as starch and protein, into smaller molecules, which release encapsulated nutrients that are more readily digested and absorbed by pigs [1,2,3,4]. Furthermore, fermentation can also reduce the immune reactivity of soy protein [5]. Fermentation metabolites, such as lactic acid and probiotics, in the fermented feed are able to suppress the proliferation of pathogens by lowering gastro-intestinal pH [6] and improving the microecology of the gut [3,7]. This was proven by a meta-analyses, which revealed that fermentation significantly increased the CP content in feed and that fermented feed significantly improved the ADG, G:F, DM digestibility, N digestibility, and energy digestibility of weaned piglets [8].

Solid fermented feed with little free water is the main form of fermented feed in China as the feeding devices in the farms are all designed for solid feed and are not suitable for liquid feed. It is well known that solid fermented feed can increase the feed intake of animals since fermentation can improve the palatability of feed by reducing the bitter and beany off-flavors of soybean meal and producing flavor substances such as lactic acid [5]. A vast literature links endocrine systems to the control of eating behavior. Three hormonal signals have been hypothesized to control hunger or satiation, cholecystokinin (CCK), leptin, and ghrelin [9]. Additionally, orexins (OX), which consist of two peptides, orexin-A (OXA) and orexin-B (OXB), derived from the proteolytic processing of a 130-amino acid precursor, preproorexin (PPOX), have also been hypothesized to promote food intake [10]. However, there are few reports demonstrating whether fermented feed-enhanced feed intake is involved in the hormone-mediated feeding regulation of animals.

The gastrointestinal tract is the main site of the digestion, absorption, and function of fermented feed, and it is also the largest hormone-producing organ in the body; these hormones play versatile roles in signal communication between the digestive tract and the central neural system [11,12,13,14]. The orexins were initially discovered in the neurons of the lateral hypothalamus[15], but orexins and cognate receptors have also been identified in numerous endocrine cells of the gastrointestinal tract [16]. Interestingly, the co-localization of ghrelin and gastrin with orexin in the same endocrine cells of the gastric glands has been identified, and this suggests that these gut peptides may collaborate in the regulation of gastric secretion, energy homeostasis, body weight, and food intake [17]. However, limited research has reported the effects of fermented feed on hormones, especially brain–gut peptide hormones in weaned pigs. This study was conducted to investigate the effects of fermented feed on the growth performance, nutrient digestibility, serum biochemical indices, serum hormones, and gene expression of brain–gut peptides and nutrient transporters in the gastric and intestinal mucosa of weaned piglets.

## 2. Materials and Methods

### 2.1. Preparation of Fermented Feedstuff (FFS)

*Lactobacillus casei* LC3 (CGMCC NO. 16132; Microbial Institute of the Chinese Academy of Sciences, Beijing, China) and *Bacillus subtilis* (CGMCC NO. 1.2428) were grown in de Man Rogosa Sharpe (MRS) broth and beef extract peptone yeast extract medium (BPY) broth at 37 °C. When the *L. casei* LC3 and *B. subtilis* content reached 2 × 10^9^ colony forming units (CFU)/mL (1–2 days), the liquid starter cultures were prepared by mixing bacterial cultures at a ratio of 5:3 (*v*/*v*) before fermentation.

Soybean meal, corn, and wheat brain were utilized as the substrates for fermentation after being milled through a 1.0 mm screen and mixing at a ratio of 5:3:2. The fermented feedstuff (FFS) was created under the following processing conditions. In brief, liquid starter culture (1% *v*/*w*) was added to the substrate. To achieve 40% initial moisture content, 50% (*v*/*w*) of water was added to the feedstuff. The fermentation was performed in multilayer polythene bags (10 kg capacity) equipped with a gas pressure opening valve at room temperature (around 25 °C) for 5 days. 

### 2.2. Experimental Design, Animals and Diets

The animal use protocols were reviewed and approved by the Animal Care and Use Committee of the Shanghai Academy of Agricultural Sciences. The experimental protocols used for research purposes in the present study were approved by the Institutional Review Board (SAASPZ0520011). A total of 320 weaned pigs (Duroc × Landrace× Yorkshire) with a similar initial body weight (7.38 ± 0.24 kg) and age (28 ± 2 d) were bred on a farm (Jinzhu Pig Farmer, Jiangsu, China) and transferred to a separate barn to conduct the experiment. At the beginning of the experiment, weaned pigs were randomly assigned to two treatment groups with eight replicates and 20 piglets per replicate on the basis of their initial BW and sex and then fed with the basal diet (unfermented feed, UFD) or the treatment diet (fermented feed, FD). 

The UFD diet was formulated to meet or exceed the nutrient requirements recommended by the NRC (2012). To construct the FD diet, 1.5% corn, 2.5% soybean meal, and 1% wheat bran in the basal diet formular were replaced with 7.5% FFS, because the extra 2.5% comes from the moisture in FFS. The ingredients and compositions of the control and treatment diet are presented in Table 1. Pigs were fed with the experimental diets throughout the 21-day experimental period after a 3-day adjustment period (replacement of the creep feed with the experiment diets). Pigs had ad libitum access to feed and water. 

### 2.3. Sampling

On day 10, pigs were deprived of feed for 12 h before blood sampling to avoid the effect of feed intake on gastro-intestinal (GI) hormone secretion. Blood samples were collected from 6 pigs per treatment from the precaval vein into nonheparinized vacuum tubes and then centrifugated at 3000× *g* for 10 min at 4 °C and stored at −20 °C until analyses for serum parameter analysis. One pig was selected randomly from each pen (6 pens for each treatment) and euthanized, and the fundic mucosal layer of the stomach, jejunum, and ileum were sampled and stored at −80 °C until analyses of the mRNA levels of target genes.

### 2.4. Growth Performance Estimation

Individual body weight (BW) was recorded on days 1 (the beginning of experimental period), 7, and 21 (the end of the experiment), and feed consumption was recorded weekly on a pen basis to obtain the average daily gain (ADG) and average daily feed intake (ADFI), respectively. Since the higher moisture content in FD resulted from the FFS administration, ADFI was calculated based on the dry matter (DM) analysis data of the diets. Then, feed conversion (FCR) was calculated using ADG and ADFI, where FCR = ADFI/ADG. 

### 2.5. Apparent Digestibility Estimation

Acid insoluble ash (AIA) was used as a digestibility indicator to determine the apparent total tract digestibility [18]. Before the experiment began, the experimental diets were sampled once and stored at −20 °C for chemical analysis. Feces were collected on day 7 to 10 from each pen (8 pens for each treatment), supplemented with 10% sulfuric acid to fix excreta nitrogen after collection, and dried in forced air (65 °C) for 72 h. The dried samples were ground and kept for AIA, dry matter (DM), crude protein (CP), and crude fiber (CF) analysis using the AOAC method [19], and phosphorus was analyzed according to the Chinese National Standard [20]. The digestibility was calculated using the following formula: digestibility (%) = (100 − A1/A × F2/F1 × 100), in which A1 represents the AIA content of the feed, A2 represents the AIA content of the feces, F1 represents the nutrient content of the feed, and F2 represents the nutrient content of the feces.

### 2.6. Plasma Biochemical Profile Detection

Plasma biochemical indices, including total protein (TP), albumin (ALB), globulin (GLB), γ-glutamyltransferase (GGT), aspartate aminotransferase (AST), alanine aminotransferase (ALT), urea, glucose (GLU), total cholesterol (TCHOC), triglyceride (TG), total bile acid (TBA), low-density leptin cholesterol (LDL-C), and high-density leptin cholesterol (HDL-C) were determined using an automatic blood analyzer (Mindray BS-420) in accordance with the manufacturer’s instructions (Shenzhen Mindray Bio-Medical Electronics Co., Ltd., Shenzhen, China).

### 2.7. Measurement of Serum Hormones

The serum levels of CCK, leptin, IGF-1, and GH were measured using commercially available ELISA kits (Shanghai Enzyme-linked Biotechnology Co., Ltd. Shanghai, China) according to the manufacturer’s instructions. Acylated ghrelin was measured using a commercially available ELISA kit (Bertin Pharma, Montigny-le-Bretonneux, France; A05401, intra-assay CV < 7.3%, inter-assay CV < 16.4%, sensitivity is 3.3 pg/mL) according to the manufacturer’s instructions. Orexin A was quantified using a commercially available ghrelin EIA kit (Phoenix Pharmaceuticals, Inc., Burlingame, CA, USA, intra-assay CV < 10%, inter-assay CV < 15%, sensitivity is 0.2 ng/mL). All samples were run in triplicate. GH and IGF-1 were assayed using commercially available ELISA kits (Shanghai Enzyme-linked Biotechnology Co., Ltd. Shanghai, China) according to the manufacturer’s instructions.

### 2.8. Extraction of RNA and cDNA Synthesis

Total RNA was extracted using an RNeasy Mini kit (QIAGEN, cat. nos. 74104) from the fundic mucosal layer of the stomach, jejunum, and ileum according to the manufacturer’s instructions. The tissues were homogenized using a FastPrep-24^®^ Homogenizer (MP Biomedicals, (Shanghai) Co., Ltd., China, Shanghai). The RNA quantity was determined using a Q5000 spectrophotometer at 260/280 nm (Quawell Technology Inc., San Jose, CA, USA). cDNA for QRT-PCR assays was synthesized from 500 mg of purified RNA using the FastKing gDNA Dispelling RT SuperMix (TIANGEN Biotech (Beijing) Co., Ltd., China, Beijing) following the manufacturer’s instructions.

### 2.9. Real-Time Fluorescence PCR for Quantification of mRNA Levels

Quantitative PCR analysis was carried out using a QuantStudio™ 6 Flex Real-Time PCR System (Applied Biosystems, Foster City, CA, USA) with TB Green^®^ Premix Ex Taq™ II (Takara Biomedical Technology (Beijing) Co., Ltd., China). cDNA was diluted 10-fold, and 2 μL of each diluted sample was added to a 20 μL reaction solution containing 10 μL of 2× master mix, 0.8 μL of each primer (10 μmol/L), and 0.4 μL of ROX II reference dye. The cycling parameters were as follows: 2 min at 95 °C, then 40 cycles of 95 °C for 15 s, annealing temperature for 30 s.

All primer sets used in this study were obtained from Sangon Biotech (Shanghai) Co., Ltd., China, Shanghai. The primers and expected product sizes are shown in Table 2. The product sizes were verified using agarose gel electrophorese. Glyceraldehyde 3-phosphate dehydrogenase (GAPDH) and tyrosine 3-monooxygenase/tryptophan 5-monooxygenase activation protein zeta polypeptide (YWHAZ) were used as reference genes, and the geometric mean of their Ct value was used as an internal control [21] to calculate the relative expression level of the target gene using the ΔΔCt method [22]. All PCR reactions were performed in triplicate, and the changes in gene expression were reported as fold increases relative to the controls.

### 2.10. Statistical Analysis

All data are shown as mean ± standard error of mean (SEM). Data processing and analysis were performed with Microsoft Excel and SPSS 13.0 software. The data were analyzed using independent-sample t tests. Significant differences were defined as *p* < 0.05.

## 3. Results

### 3.1. The Effects of FFS on Growth Performance of Pigs

All pigs were healthy and grew well throughout the entire experimental period. As shown in Table 3, our results showed that the FD group had higher ADFI and ADG during the first week (*p* < 0.01 and *p* < 0.05), during the last two weeks (*p* < 0.05 and *p* < 0.01), and over the entire three-week period (*p* < 0.05 and *p* < 0.01) compared with the UFD group. At the end of the experiment, the FD group had a higher BW compared with the UFD group (*p* < 0.05). Moreover, the FCR of pigs was improved by FFS replacement during the last two weeks (*p* < 0.05), while there were no significant differences during the first week and over the entire three-week period (*p* > 0.05).

### 3.2. The Effects of FFS on the Apparent Nutrient Digestibility of Pigs

As shown in Table 4, compared with the UFD group the FD group significantly increased the phosphorus digestibility (*p* < 0.05) and tended to increase the CP but decrease CF digestibility (*p* < 0.1). However, no significant difference was observed in the DM digestibility between these two groups.

### 3.3. The Effects of FFS on the Serum Biochemical Parameters of Pigs

The effects of feeding pigs with fermented feed on the plasma biochemical parameters are presented in Table 5. The results showed that, compared with the UFD group, the FD group tended to have increased concentrations of serum AST and TBA (*p* < 0.1). However, no significant differences were observed in the concentration of serum glucose, urea, phosphorus, ALB, GLB, GGT, AST, TG, TCHO, LDL-C, or HDL-C levels between the UFD and FD groups (*p* > 0.05). 

### 3.4. The Effects of FFS on the Serum Hormones Associated with Feed Intake and Growth of Pigs

To further elucidate the mechanism by which the fermented feedstuff promoted the feed intake and weight gain in pigs, the concentrations of serum hormones associated with feed intake and growth were measured in the present study. As shown in Figure 1, compared with the UFD group, the FD group had significantly increased concentrations of serum orexin (*p* < 0.05) and tended to have increased concentrations of serum IGF-1 and CCK (*p* < 0.1). However, no significant differences were observed in the serum concentrations of GH, leptin, or acylated ghrelin between the UFD and FD groups (*p* > 0.05). 

### 3.5. The Effects of FFS on the Gene Expressions of Nutrient Transporters of Pigs

As shown in Figure 2a, the genes of excitatory amino acid transporter 3 (EAAT3), sodium–glucose linked transporter1 (SGLT1), and peptide transporter1 (PepT1) were all expressed in the ileum and jejunum of pigs, but these genes were more highly expressed in the ileum than in the jejunum when calculated in terms of relative expression (2^ΔΔCt^ method). Their abundances in the ileum were approximately 50-, 9-, and 3-fold higher than in the jejunum, respectively. The statistical results showed that, compared with the UFD group, the FD group had significantly increased relative expression of SGLT1 and PepT1 in the jejunum (*p* < 0.05, Figure 2b), but not in the ileum (*p* > 0.05, Figure 2c). However, there were no significant differences in the relative expressions of the gene EAAT3 between the UFD and FD groups in either the ileum or the jejunum (*p* > 0.05).

### 3.6. The Effects of FFS on the Gene Expressions Associated with Feed Intake of Pigs

As the gene expression is associated with the feed intake of pigs, our results showed that the genes PPOX, ghrelin, and gastrin were all expressed in the gastric fundus, jejunum, and ileum of pigs (Figure 3a). Among these three tissues, PPOX and ghrelin were more highly expressed in the gastric fundus (Figure 3a), and their abundances were approximately 500- and 150,000-fold higher than in the jejunum, and approximately 17- and 120,000-fold higher than in the ileum, respectively. In addition, the gastrin gene was more highly expressed in the gastric fundus and ileum than in the jejunum.

The statistical results showed that (Figure 3), compared with the UFD group, the FD group had significantly higher prepro-orexin (PPOX) gene expression in the gastric fundus, jejunum, and ileum (*p* < 0.01) and higher ghrelin gene expression in the gastric fundus (*p* < 0.1). However, no significant differences in FFS were observed in the relative expressions of gene gastrin in the gastric fundus, jejunum, or ileum of pigs (*p* > 0.05).

### 3.7. The Effects of FFS on the Gene Expressions of IGF-1 and IGFR in the Intestinal Mucosa of Pigs

As shown in Figure 4a, the genes of IGF1 and IGFR were more highly expressed in the ileum than those in the jejunum when calculated in terms of relative expression (2^ΔΔCt^ method). Their abundances were approximately 50-fold higher in the ileum than in the jejunum. The statistical results showed that, compared with the UFD group, the FD group had significantly higher IGF1 and IGFR gene expression in the jejunum of pigs (*p* < 0.01, Figure 4b) but not in the ileum (*p* > 0.05, Figure 4c).

## 4. Discussion

In order to eliminate the influence of feed ingredients on the experimental results and reduce the loss of animal protein and amino acid additives caused by fermentation, FFS was used in this study to replace the same amount of feedstuff. Similar to the findings of other researchers [3,8,27,28,29,30,31], in the present study, pigs in the FD group had significantly higher ADFI, ADG, FCR, and phosphorus digestibility and higher CP but lower CF digestibility compared with pigs in the UFD group. Koo et al. [2] reported that inoculating wheat with *L. buchneri* was more beneficial than inoculating it with *L. plantarum* in terms of energy and mineral digestibility, but no differences in amino acid digestibility were observed between the inoculants. Hu et al. [32], reported that a solid-state fermented diet increased the apparent total tract nutrient digestibility of CF, CP, neutral detergent fiber (NDF), acid detergent fiber (ADF), eight amino acids (Trp, Asp, Gly, Cys, Val, Met, Ile, and Leu), total essential amino acids (EAA), total non-essential amino acids (NEEA), and total amino acids (TAA). The higher phosphorus and CP digestibility of the FFS group in the present study might be caused by the microbial degradation of antinutritional factors in the feed, such as cell wall components and phytates, and the breakdown of the large protein molecules into smaller molecules. Meanwhile, fermentable fibers were utilized during fermentation, resulting in the lower CF digestibility of the FFS group. Therefore, our results indicated that dietary supplementation with FFS was able to enhance the digestion of nutrients in the digestive tract, promote feed intake, and thus improve the growth performance of weaned pigs in the present study.

Evidence showed that blood biochemical parameters reflected the comprehensive functions of body organs and nutritional metabolism, which could indirectly indicate the health status of pigs [26]. Hence, to further understand the overall impacts of FFS on the host, we detected the changes in serum biochemical parameters. In the study of Hu et al. [32], their results showed that a solid-state fermented diet increased the serum concentrations of HDL-C, ALB, phosphors, and glucose but reduced the serum concentrations of TC, TG, and LDL. However, no significant effect of FFS was observed in the serum TP, urea, or phosphorus, although higher CP and phosphorus digestibility were observed in the FD group. Moreover, no significant effect of FFS was observed in the other serum biochemical profiles. Similarly, Ding et al. [31] reported that the dietary supplementation of 5% fermented soybean meal improved ADG, ADFI, and FCR, but had no effect on the blood profile. Another study by Wang et al. [28] also reported that a diet containing 6% fermented soybean meal (FSBM) improved ADG and ADFI in weaned pigs but had no effect on the nutrient digestibility and serum urea N concentration. In general, dietary supplementation with FFS did not alter the serum biochemical parameters of pigs, which confirmed the feasibility and safety of the use of FFS in pig diets to a certain extent in the present study.

To further reveal the possible mechanism of fermented diets improving the growth performance of weaned pigs, we quantified the gene expression of nutrient transporters in the jejunum and ileum of piglets, which are responsible for the absorption of amino acid, small peptides, and glucose. PepT1 is responsible for the transport of dipeptides, tripeptides and their analogues, all of which originate from the digestion of proteins [33]; EAAT3 is an epithelial type high-affinity anionic amino acid transporter [25]; and SGLT1 is pivotal for intestinal glucose absorption and glucose-dependent incretin secretion [34]. In the present study, the abundance of EAAT3, SGLT1, and PepT1 in the ileum was approximately 50-, 9-, and 3-fold higher than that in the jejunum, respectively. However, compared with pigs in UFD, pigs in FD only had a significantly higher SGLT1 and PepT1 gene expression in the jejunum—not in the ileum. The results of this study indicated that FFS only improved the gene expression of nutrient transporters in the jejunum, while the nutrient transport and absorption mainly occurred in the ileum. This also explains why there were no significant differences in the serum biochemical parameters between the two groups of pigs in this study.

It is well known that fermented diets can increase the feed intake of animals, which is important for improving animal growth. Indeed, our abovementioned result showed that dietary supplementation with FFS significantly increased the ADFI of weaned pigs in the present study. In addition to improving palatability, whether fermented diets could promote feeding by regulating endocrine hormones is rarely reported and thus attracted our attention. Xin et al. [30] reported that feeding piglets with liquid fermented feed significantly elevated the level of serum ghrelin in weaned piglets but tended to decrease the level of serum leptin. Ghrelin is an acylated peptide that plays an important role in energy homeostasis, body weight control, and food intake [35]. A previous study showed that weaned pigs intravenously infused with human ghrelin showed increased eating times and weight gain compared with saline-infused controls [36]. In the present study, no significant effect was observed on the serum level of leptin or acylated ghrelin, and the level of serum CCK in the FD group had a tendency to increase. The effects of leptin and CCK on appetite include a decrease in feed intake in animals and humans. Specifically, the secretion of CCK is stimulated by fat and protein in the chyme, which elicits multiple effects on the gastrointestinal system, including the regulation of gastric emptying, gut motility, contraction of the gallbladder, pancreatic enzyme secretion, and gastric acid secretion [37]. The exogenous administration of CCK into the peritoneal cavity (ip) of rats dose-dependently reduced meal size [38]. However, if CCK is administered more than 15 min before animals begin eating, it has no effect on meal size. Furthermore, behavioral experiments indicate that CCK is a short-term, meal-reducing signal [39]. Although the effects of exogenous CCK are brief, acting within the time of an ongoing meal, CCK also appears to interact with long-term signals of energy balance, such as leptin [40]. However, numerous studies have reported that circulating leptin concentrations are strongly and positively correlated with adiposity (positive energy balance) [9,41]. With long-term positive energy balance, leptin steadily increases in serum to reflect increased adipose tissue mass. Conversely, leptin concentrations in serum and adipose tissue rapidly and profoundly decrease as a result of food deprivation and negative energy balance. However, since fat deposition is limited in weaning pigs, no significant difference was observed in the serum leptin levels between the two groups, though fermented feed significantly improved the ADFI and ADG of the weaned pigs in our study. 

Orexin, a hypothalamic neuropeptide, has been confirmed to promote food intake [42]. Dall’Aglio et al. [17] reported that orexins (OXA and OXB) and their cognate receptors (OX1R and OX2R) were detected only in the gastric glands in pigs and were co-localized with ghrelin and gastrin in the same endocrine cells. Similarly, in the present study, PPOX and ghrelin were most highly expressed in the gastric fundus, and their abundances were approximately 500- and 150,000-fold higher than in the jejunum and approximately 17- and 120,000-fold higher in the ileum, respectively. In the present study, the results showed that the serum orexin level was increased and the prepro-orexin (PPOX) mRNA level was increased in the gastric fundus, jejunum, and ileum mucosa of weaned pigs fed with FD. Nevertheless, we did not detect the mRNA of OXR1 and OXR2 in the gastric fundus, jejunum, or ileum of pigs, using the same primers, which may indicate that FFS induced greater orexin secretion to the circulatory system and caused the receptors in the hypothalamus to regulate feeding. Interestingly, in addition to promoting feed intake, orexin can promote a number of phenomena involved in successful foraging, including food anticipatory locomotor behavior, olfactory sensitivity, visual attention, spatial memory, and mastication [10]. Unfortunately, we did not observe whether there was a difference in feeding behavior between the two groups, and hence our future studies should focus on this aspect. Nevertheless, the above results confirmed that the increased feed intake after dietary supplementation with FFS was associated with increased serum orexin levels and prepro-orexin (PPOX) mRNA in the digestive tract of pigs.

Feed intake and growth are two interrelated physiological processes. As we mentioned above, our results showed that dietary supplementation with FFS enhanced the digestion and absorption of nutrients and increased the ADFI of pigs, and this was one of the aspects promoting animal growth in the present study. As it is well known that hormones also play an important role in the growth and development of animals, we also detected the serum GH and IGF-1, and the gene expressions of IGF-1 and IGFR in intestinal mucosa in the present study. It was reported that piglet weight was significantly positively correlated with the serum IGF-I level, but not with the serum GH level [43]. Similarly, in the present study, an increased tendency of serum IGF-1 was observed in the FD group, but there was no significant difference in the serum GH level between these two groups. In accordance with this result, the analysis of real-time PCR showed that, compared with the UFD group, the FD group had significantly higher IGF-1 and IGFR gene expression in the jejunum. Hence, the elevated serum IGF-1 level and increased relative expression of IGF-1 and IGFR in the intestinal mucosa also contributed to the growth of weaned pigs fed fermented feed in the present study.

## 5. Conclusions

Dietary supplementation with fermented feedstuff improved ADFI, ADG, FCR, and phosphorus digestibility and tended to increase the CP and CF digestibility in weaning pigs. Pigs fed fermented feed had a higher serum orexin level and up-regulated expression of the PPOX gene in the gastric fundus, jejunum, and ileum mucosa, while the expression of IGF-1 and IGFR genes in the jejunum was comparable to that of pigs fed an unfermented diet. Results indicated that dietary supplementation with fermented feedstuff improved the growth performance of weaned pigs by increasing orexin, IGF-1 and IGFR levels.

## Figures and Tables

**Figure 1 animals-12-01329-f001:**
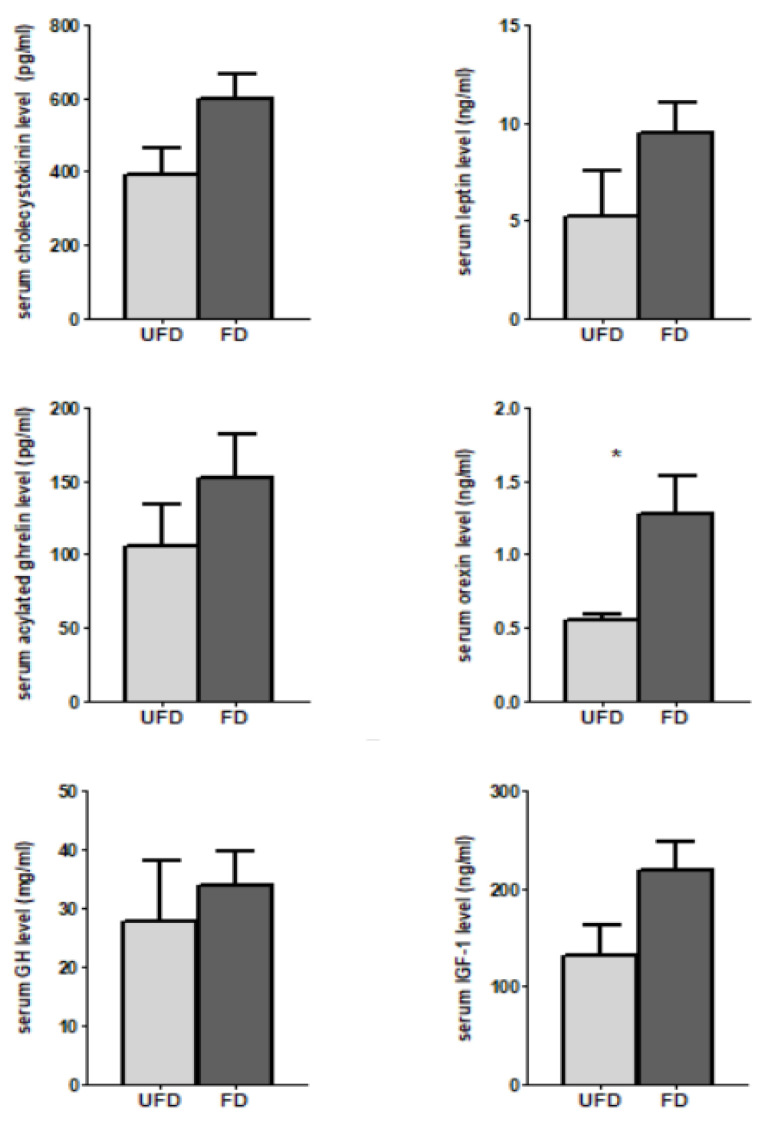
Effects of fermented feedstuff (FFS) on serum hormones associated with food intake and growth in weaned piglets. The serum levels of cholecystokinin (CCK), leptin, IGF-1, and GH were measured by commercially available ELISA kits (Shanghai Enzyme-linked Biotechnology Co., Ltd. Shanghai, China). Orexin A was quantified by a commercially available ghrelin EIA kit (Phoenix Pharmaceuticals, Inc., CA, USA, intra-assay CV < 10%, inter-assay CV < 15%, sensitivity is 0.2 ng/mL). Acylated ghrelin was measured by a commercially available ELISA kit (Bertin Pharma, Montigny-le-Bretonneux, France; A05401, intra-assay CV < 7.3%, inter-assay CV < 16.4%, sensitivity is 3.3 pg/mL). The results are shown as mean ± S.E.M. UFD: unfermented feed group; FD: fermented group. Bars with one star denote a significant difference (*p* < 0.05).

**Figure 2 animals-12-01329-f002:**
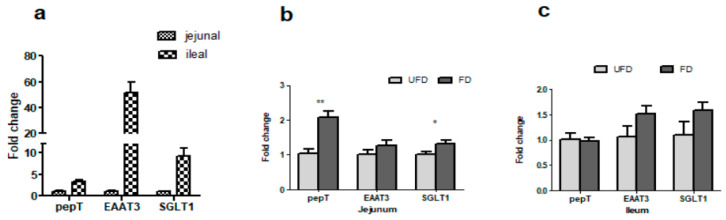
Relative gene expression of nutrient transport peptide genes in porcine jejunum and ileum (**a**) and effects of fermented feedstuff (FFS) on the expression of the two genes in the jejunum (**b**) and ileum (**c**) of weaned piglets. The mRNA expression was measured by real-time PCR with 5 mg of total mRNA. Glyceraldehyde 3-phosphate dehydrogenase (GAPDH) and tyrosine 3-monooxygenase/tryptophan 5-monooxygenase activation protein zeta polypeptide (YWHAZ) were used as reference genes, and the geometric mean of their Ct value was used as an internal control. The results of the feed change are shown as mean ± S.E.M. EAAT3: excitatory amino acid transporter3; PepT1: peptide transporter 1; SGLT1: sodium–glucose linked transporter 1; UFD: unfermented feed group; FD: fermented group. Bars with one star denote a significant difference (*p* < 0.05). Bars with two stars mean very significantly different *(p* < 0.01).

**Figure 3 animals-12-01329-f003:**
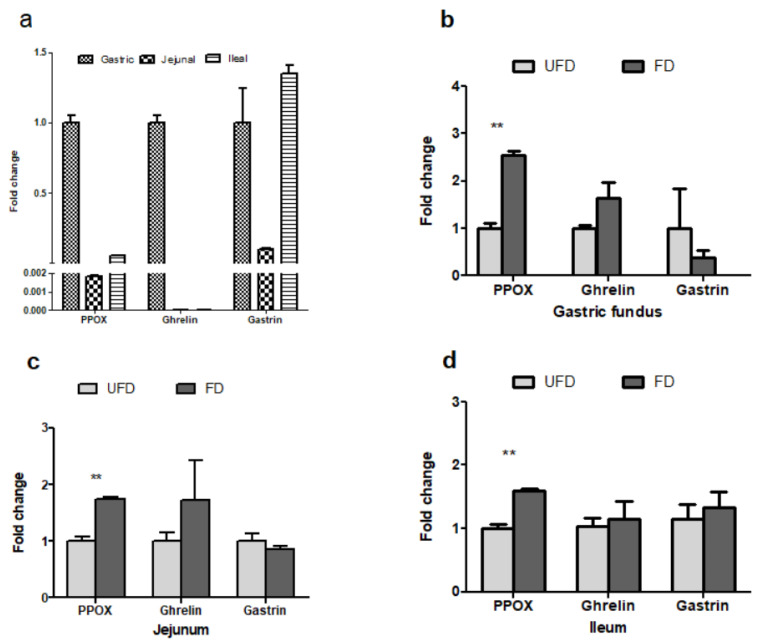
Relative gene expression of PPOX, ghrelin, and gastrin in porcine gastric fundus, jejunum, and ileum mucosa (**a**) and effects of fermented feedstuff (FFS) on the expression of the three genes in the gastric fundus (**b**), jejunum (**c**), and ileum (**d**) of weaned piglets. The mRNA expression was measured by real-time PCR with 5 mg of total mRNA. Glyceraldehyde 3-phosphate dehydrogenase (GAPDH) and tyrosine 3-monooxygenase/tryptophan 5-monooxygenase activation protein zeta polypeptide (YWHAZ) were used as reference genes, and the geometric mean of their Ct value was used as an internal control. The results of the feed change are shown as mean ± S.E.M. UFD: unfermented feed group; FD: fermented group. Bars with two stars denote a significant difference *(p* < 0.01).

**Figure 4 animals-12-01329-f004:**
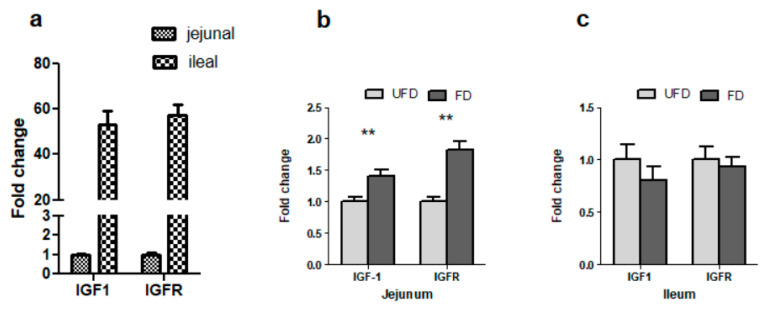
Relative gene expression of IGF-1 and IGFR in porcine jejunum and ileum mucosa (**a**) and effects of fermented feedstuff (FFS) on the expression of the two genes in the jejunum (**b**) and ileum (**c**) of weaned piglets. The mRNA expression was measured by real-time PCR with 5 mg of total mRNA. Glyceraldehyde 3-phosphate dehydrogenase (GAPDH) and tyrosine 3-monooxygenase/tryptophan 5-monooxygenase activation protein zeta polypeptide (YWHAZ) were used as reference genes, and the geometric mean of their Ct value was used as an internal control. The results of the feed change are shown as mean ± S.E.M. UFD: unfermented feed group; FD: fermented group. Bars with two stars denote a highly significant difference *(p* < 0.01).

**Table 1 animals-12-01329-t001:** Composition and nutrient levels of the control and treatment diet (% as fed).

Ingredients (%)	UFD ^d^	FD ^d^
Corn	61.56	60.06
Soybean meal	13.8	11.3
Wheat bran	1	0
FFS ^a^	0	7.5
Fish meal	2	2
Extruded soybean	10	10
Soybean oil	2.5	2.5
Soybean protein	2.5	2.5
Sugar	2	2
Acidifier	0.4	0.40
Sodium chloride	0.38	0.38
Calcium carbonate	0.7	0.7
Calcium phosphate	1.3	1.30
Lysine HCl	0.6	0.6
L-Threonine	0.13	0.13
DL-Methionine	0.1	0.1
Tryptophan	0.03	0.03
Premix ^b^	1	1
Nutrition level ^c^		
ME (Mcal/kg)	3.51	3.51
Crude protein%	18.74	18.98
Lysine%	1.57	1.57
Calcium%	0.74	0.74
Phosphorus%	0.59	0.59

^a^ FFS (fermented feedstuff) comprised 1.5% corn, 2.5% soybean meal, 1% wheat bran, and 2.5% water. ^b^ Premix supplied per kilogram diet: 35 mg of Mn, 100 mg of Fe, 90 mg of Zn, 16.5 mg of Cu, 0.3 g of I, 0.3 mg of Se, 11,000 IU of Vitamin A, 1503 IU of Vitamin D3, 44.1 IU of Vitamin E, 4 mg of menadione, 5.22 mg of riboflavin, 20 mg of pantothenic acid, 0.3 mg of folic acid, and 1 g of choline. ^c^ Nutrition level value was calculated excluding crude protein. ^d^ UFD: unfermented feed group; FD: fermented group.

**Table 2 animals-12-01329-t002:** List of primers used for quantitative PCR.

Gene Name	Primer Sequence (5′–3′)	Product Length (bp)	Ref
PPOX ^1^	For.: TTCCTGGAGACCCCACTCTG	101	[23]
	Rev.: GGCAGCAACAGTAGCAGAAG		
Ghrelin	For.: GCTCCTCATGGCAGACTT	92	
	Rev.: CTGGCTTCTTGGACTCCTT		
Gastrin	For.: TCCTCAGCACTGCGGCGG	87	
	Rev.: ATGGAGGAGGAAGAAGAAGC		[24]
IGF-1	For.: GCACATCACATCCTCTTCGC	164	
	Rev.: ACCCTGTGGGCTTGTTGAAA		
IGF-1R	For.: CTGTGGGGGCTCCTGTTTTT	200	
	Rev.: GTGAGCTTGGGAAAGCGGTA		
EAAT3	For.: GGCACCGCACTCTACGAAGCA	177	[25]
	Rev.: GCCCACGGCACTTAGCACGA		
PepT1	For.: GGTTTAGGCATCGGAGTAAGAAGT	156	[26]
	Rev.: GGTCAAACAAAGCCCAGAACAT		
SGLT1	For.: GGCTGGCGAAGTATGGTGT	153	
	Rev.: ACAACCACCCAAATCAGAGC		
YWHAZ	For.: TGATGATAAGAAAGGGATTGTGG	203	
	Rev.: GTTCAGCAATGGCTTCATCA		
GAPDH	For.: GTTCCAGTATGATTCCACCCACGGCA	147	
	Rev.: TGCCAGCCCCAGCATCAAAGGTAGAA		

^1^ PPOX: prepro-orexin; IGF-1: insulin-like growth factor 1; IGF-1R: insulin-like growth factor 1 receptor; EAAT3: excitatory amino acid transporter 3; PepT1: peptide transporter 1; SGLT1: sodium–glucose linked transporter 1; YWHAZ: tyrosine 3-monooxygenase/tryptophan 5-monooxygenase activation protein zeta polypeptide; GAPDH: glyceraldehyde 3-phosphate dehydrogenase (GAPDH).

**Table 3 animals-12-01329-t003:** Effect of dietary supplementation with FFS on growth performance in weaned pigs.

Phase	Item	UFD	FD	SEM	*p* Value
BW, Kg	Initial	7.41	7.35	0.35	0.912
	d 7	8.49	8.61	0.38	0.319
	d 21	11.68	12.14	0.48	0.009
d 1–7	ADFI ^1^,g/d	174	188	6.10	0.002
	ADG,g/d	134	161	4.04	0.017
	FCR	1.29	1.19	0.06	0.146
d 8–21	ADFI,g/d	387	418	18.09	0.026
	ADG,g/d	262	327	15.58	0.006
	FCR	1.48	1.31	0.06	0.040
Overall (d 1–21)	ADFI,g/d	312	338	8.16	0.028
	ADG,g/d	216	245	9.55	<0.001
	FCR	1.45	1.38	0.03	0.120

^1^ ADFI: average daily feed intake was adjusted based on the dry matter (DM); ADG: average daily gain; FCR: feed conversion. Values are means and pooled S.E.M. (n = 8); UFD: unfermented feed group; FD: fermented group, BW: body weight; ADFI: average daily feed intake; ADG: average daily gain; FCR: feed conversion.

**Table 4 animals-12-01329-t004:** Effect of dietary supplementation with FFS on the apparent nutrient digestibility in weaned pigs.

Item	UFD	FD	SEM	*p* Value
Dry matter (%)	83.85	84.05	0.68	0.847
Crude protein (%)	76.65	79.61	0.98	0.054
Crude fiber (%)	64.56	56.66	2.60	0.076
Phosphorus (%)	42.59	53.46	2.74	0.025

Values are means and pooled S.E.M. (n = 8); UFD: unfermented feed group; FD: fermented group.

**Table 5 animals-12-01329-t005:** Effects of dietary supplementation with FFS on the serum biochemical parameters in weaned pigs.

Item	Unit	UFD	FD	SEM	*p* Value
TP	g/L	51.56	49.77	1.55	0.443
ALB	g/L	30.69	29.41	0.90	0.339
GLB	g/L	20.86	20.43	1.44	0.839
ALB/GLB		1.49	1.50	0.12	0.949
Glucose	mmol/L	6.41	6.77	0.50	0.629
Urea	mmol/L	5.89	5.34	0.39	0.333
Phosphorus	g/L	2.27	2.31	0.07	0.716
GGT	U/L	43.29	41.86	4.35	0.832
AST	U/L	133.40	180.30	16.29	0.077
ALT	U/L	48.00	50.00	4.47	0.757
TG	mM	1.58	1.66	0.07	0.481
TCHO	mmol/L	1.58	1.80	0.10	0.141
TBA	μmol/L	19.47	31.32	4.57	0.093
LDL-C	mM	0.80	0.94	0.06	0.137
HDL-C	mM	0.60	0.66	0.05	0.366

TP: total protein; ALB: albumin; GLB: globulin; GGT: γ -glutamyltransferase; AST: aspartate aminotransferase; ALT: alanine aminotransferase; TG: triglyceride; TCHO: total cholesterol; TBA: total bile acid; LDL-C: low-density leptin cholesterol; HDL-C: high-density leptin cholesterol. Values are means and pooled S.E.M. (n = 6); UFD: unfermented feed group; FD: fermented group.

## Data Availability

The data supporting the conclusions of this article are all available.

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
