# Peer review of "Supplementation with Fermented Feedstuff Enhances Orexin Expression and Secretion Associated with Increased Feed Intake and Weight Gain in Weaned Pigs"

_animals, 2022, doi:10.3390/ani12101329_

Round 1
Reviewer 1 Report
“Supplement of Fermented Feed Stuff Enhances Orexin Expression and Secretion Associated with Increased Feed Intake and Weight Gain in Weaned Pigs”
This MS needs to prove the certificate number or series of ethics approval for Animal care and use and rewrite the simple summary and introduction.
Simple Summary: too shot to explain in the overview of this paper and due to make understanding for this publication. The author has to rewrite more.
Introduction:
- The introduction does not have the consistency of the content. The introduction has to provide sufficient background about the genes that are related to the topic or subject, i.e. Orexin or background of the gene expression of brain-gut peptides that are related to this study. That still does not reflect of topic and content.
- Line 50: “Large molecules of starch and protein of feed….” Are you sure this right explanation from the reference? Starch is quite easy to digest; In the ref. they talk about polysaccharides.
Material and methods:
- The author has to write the certificate number or series of ethics approval. The information is not sufficient.
- Line 100: delete “(UPD)” because mention already in line 98. Same as Line 101 delete “(FD)”
- -Table 1. “Composition and nutrient levels of the basal diet (% as fed).” This table show composition of all treatment, not only the basal diet. The author has to rewrite.
- And “NDF” is the wrong abbreviation.
- The number of the data was not consistent. The digit was not the same in every value
- Line 122 : “g” have to italics. Every position of “g” which mentions force from centrifuge needs to write in italics.
- Line 132: “Then, feed conversion (FC)” In general, we use “feed conversion ratio (FCR)” may be easy to understand.
- The apparent total tract digestibility. The author has to provide the reference. Do this experiment use maker as chromium oxide?
Result:
- All “P” that shows P-value have to write in italics.
- Need to rewrite the format of Table 3, make it clear in the line of data
Author Response
Simple Summary: too shot to explain in the overview of this paper and due to make understanding for this publication. The author has to rewrite more.
Answer:According to the reviewer’s comment, we have rewritten the simple summary in the manuscript, as follows:
In the weaned pigs, how to improve the health status and also enhance the growth performance of piglets under weaning stress through nutritional measures is an important issue that needs to be solved. Supplemented of fermented feed stuff improved the feed intake and growth performance of weaned pigs, but the exact mechanism remains unclear. Hence, the present study evaluated the effects of fermented feed stuff on the performance and gastrointestinal hormones involved in feed intake and growth in weaned pigs. The results of our study showed that the dietary supplementation with fermented feed stuff improved growth performance of weaned pigs by increased orexin, IGF-1 and IGFR levels.
Introduction:
- The introduction does not have the consistency of the content. The introduction has to provide sufficient background about the genes that are related to the topic or subject, i.e. Orexin or background of the gene expression of brain-gut peptides that are related to this study. That still does not reflect of topic and content.
Answer:Thanks for the reviewer’s valuable comment. According to the reviewer’s comment, we have revised the introduction.
- Line 50: “Large molecules of starch and protein of feed….” Are you sure this right explanation from the reference? Starch is quite easy to digest; In the ref. they talk about polysaccharides.
Answer:Thanks for the reviewer’s valuable comment. To correspond with “protein”, we use “carbohydrate” instead of “starch”.
Material and methods:
- The author has to write the certificate number or series of ethics approval. The information is not sufficient.
Answer:Thanks for the reviewer’s valuable comment. The certificate number has been added.
- Line 100: delete “(UPD)” because mention already in line 98. Same as Line 101 delete “(FD)”
Answer:Thanks for the reviewer’s valuable comment. It has been corrected.
- -Table 1. “Composition and nutrient levels of the basal diet (% as fed).” This table show composition of all treatment, not only the basal diet. The author has to rewrite.
- And “NDF” is the wrong abbreviation.
- The number of the data was not consistent. The digit was not the same in every value
Answer:Thanks for the reviewer’s valuable comment. It has been corrected as follows: “To make the treatment FD diet (FD), 1.5% corn, 2.5% soybean meal and 1% wheat bran in the basal diet formular were replaced with 7.5% FFS, because the extra 2.5% come from the moisture in FFS.”
- Line 122 : “g” have to italics. Every position of “g” which mentions force from centrifuge needs to write in italics.
Answer:Thanks for the reviewer’s valuable comment. It has been corrected.
- Line 132: “Then, feed conversion (FC)” In general, we use “feed conversion ratio (FCR)” may be easy to understand.
Answer:Thanks for the reviewer’s valuable comment. It has been corrected.
- The apparent total tract digestibility. The author has to provide the reference. Do this experiment use maker as chromium oxide?
Answer: Thanks for the reviewer’s valuable comment. We have added the reference (ref. 13). The review of Sales (2003) had shown AIA was a reliable internal marker for determination of fecal digestibility in animal. AIA is naturally present in feed and does not need to be added in the experiment.
Reviewer 2 Report
The authors have reported a study that sought to evaluate fermented diets vs unfermented diets. The work is not novel and does not add any new information to the scientific literature in relation to the benefits of feed fermentation in weanling pigs. But also, very keys questions arise from the presentation in the manuscript which need to be addressed before consideration for publication.
- With the inclusion of 2.5% water in the formulation through the FFP, how did you reconcile the dry matter intake of the weaned pigs? Are these reported feed intake values based on ‘as is’ or DM basis? You need to clarify this and explain how this impacts the blood biochemical levels.
- The acid-insoluble ash was used to determine digestibility, however, in the diet, no additional AIA was included. Is it to assume that only the inherent AIA in the diets was used to predict digestibility?
- There are output parameters that were measured but no justifications were made for that analysis.

Author Response
- With the inclusion of 2.5% water in the formulation through the FFP, how did you reconcile the dry matter intake of the weaned pigs? Are these reported feed intake values based on ‘as is’ or DM basis? You need to clarify this and explain how this impacts the blood biochemical levels.
Answer: Thanks for the reviewer’s valuable comment. In our study, in order to eliminate the influence of feed composition on the experimental results, the feed stuff we used in the FD diet and the basic diet is the same. To make the FD diet, 1.5% corn, 2.5% soybean meal and 1% wheat bran in the basal diet formular were replaced with 7.5% FFS, since 7.5% FFS was made from 1.5% corn, 2.5% soybean meal and 1% wheat bran and 2.5% water. These descriptions were based on “as is”, if we changed the diet composition to “DM basis”, the composition of basal diet and FFS diets were the same except for the levels of corn, soybean meal, and wheat bran. We did not add water to the the basic diet, because the extra 2.5% of moisture was negligible compared to the amount of water consumed by pigs and had no significant physiological effects on pigs. However, to be fair, the feed intake values were based on the “DM basis”.
- The acid-insoluble ash was used to determine digestibility, however, in the diet, no additional AIA was included. Is it to assume that only the inherent AIA in the diets was used to predict digestibility?
Answer: Thanks for the reviewer’s valuable comment. We have added the reference (ref. 13). The review of Sales (2003) had shown AIA was a reliable internal marker for determination of fecal digestibility in animal. AIA is naturally present in feed and does not need to be added in the experiment.
- There are output parameters that were measured but no justifications were made for that analysis.
Answer: Thanks for the reviewer’s valuable comment. We have revised the discussion and deleted “the pigs in the FD group had higher plasma ALT and TBA level compared with the pigs in the UFD group in present study”, because only increase tendency was observed on them (p <0.1) in the study. Excessive discussion on the effects of FFS on blood biochemical parameters which had little relate with other data will distracts from the focus of this paper.
Round 2
Reviewer 1 Report
This manuscript is now improved
Author Response
The introduction has to provide sufficient background about the genes that are related to the topic or subject, i.e. Orexin or background of the gene expression of brain-gut peptides that are related to this study. That still does not reflect of topic and content.
Answer: On line 72 and line 81, reference 9, 10, 15 and 16 were cited in the introduction, to provide background about orexin and other hormonal signals.
Reviewer 2 Report
Second review
- In the first review on Line 134 I asked you to indicate the formula and calculation or the adjusted DM. Please indicate that or point to the line #
- I am not satisfied with the response on AIA. Indeed, the dietary ingredients have AIA and measuring AIA in diets and or fecal matter for digestibility is common, however, there could be variations in the AIA content based on the experimental treatments. That is why measured amounts celite is added to diets as a marker and measured in feces to determine digestibility. Therefore, I need to know why in your case no dietary AIA was added and why you think this might not have any effects, especially considering you fed dietary with differing DM.
- On Line 139 I asked for clarification for the number of pens from which fecal samples were collected, please address this.
- You mentioned on L204 under statistical analysis that the study was a randomised complete block design. My question was” what was the block’ did you block by sex, weights etc.?? please indicate. Even in the updated version of the paper, you did not indicate anymore the experimental design. Analysing as a t-test is not appropriate for the study design as reported.
- Overall, I see improvements in the revised version but please address the above concerns for clarity and repeatability.
- On Line 352, you had mentioned that the ALT and TBA levels in plasma was higher in the FD group than the UFD group. No explanation was given for this observation. Since this statement appears in the discussion section and not the results section, it warrants an explanation and what it might mean for the overall study observations and conclusions. I would expect to see that addressed.
Author Response
1. In the first review on Line 134 I asked you to indicate the formula and calculation or the adjusted DM. Please indicate that or point to the
Answer: In the first review on Line 134 “ADFI was adjusted based on the dry matter (DM) of the diets” was corrected as “ADFI was calculated based on the dry matter (DM) analysis data of the diets” on line 145. DM has not been adjusted. We use DM data instead of the raw dada to calculate ADFI.
2. I am not satisfied with the response on AIA. Indeed, the dietary ingredients have AIA and measuring AIA in diets and or fecal matter for digestibility is common, however, there could be variations in the AIA content based on the experimental treatments. That is why measured amounts celite is added to diets as a marker and measured in feces to determine digestibility. Therefore, I need to know why in your case no dietary AIA was added and why you think this might not have any effects, especially considering you fed dietary with differing DM.
Answer: The determination and calculation of digestibility are based on dried samples. In this experiment, the feed stuffs of the two feeds are the same, therefore the AIA content in the dried samples was also the same, although the moisture was different. We did not add external indicators in this study to avoid extra errors due to uneven mixing. Thanks for the reviewer’s valuable comment. We'll pay more attention to indicators when we do digestibility tests.
3. On Line 139 I asked for clarification for the number of pens from which fecal samples were collected, please address this.
Answer: Thanks for the reviewer’s valuable suggestion. The number of pens has been add on line 152.
4. You mentioned on L204 under statistical analysis that the study was a randomised complete block design. My question was” what was the block’ did you block by sex, weights etc.?? please indicate. Even in the updated version of the paper, you did not indicate anymore the experimental design. Analysing as a t-test is not appropriate for the study design as reported.
Answer: As the descried on line 110 “At the beginning of the experiment, weaned pigs were randomly assigned to two treatment groups with eight replicates and 20 piglets per replicate on the basis of their initial BW and sex, fed with basal diet (unfermented feed, UFD) and treatment diet (fermented feed, FD)”, the study was a completely random design with two groups. We made the initial BW and sex ratio similar between each group. Therefore, there is no block. Because there were two groups, the data were analyzed independent-sample t test (L218).
5. Overall, I see improvements in the revised version but please address the above concerns for clarity and repeatability.
Answer: Thanks for the reviewer’s valuable comment. We’ll try our best.
6. On Line 352, you had mentioned that the ALT and TBA levels in plasma was higher in the FD group than the UFD group. No explanation was given for this observation. Since this statement appears in the discussion section and not the results section, it warrants an explanation and what it might mean for the overall study observations and conclusions. I would expect to see that addressed.
Answer: Thanks for the reviewer’s valuable comment. We have revised the discussion and deleted “the pigs in the FD group had higher plasma ALT and TBA level compared with the pigs in the UFD group in present study”, because only increase tendency was observed on them (p <0.1) in the study. Excessive discussion on the effects of FFS on blood biochemical parameters which had little relate with other data will distracts from the focus of this paper.
